# The Heterogeneity of Skewness in T2W-Based Radiomics Predicts the Response to Neoadjuvant Chemoradiotherapy in Locally Advanced Rectal Cancer

**DOI:** 10.3390/diagnostics11050795

**Published:** 2021-04-28

**Authors:** Francesca Coppola, Margherita Mottola, Silvia Lo Monaco, Arrigo Cattabriga, Maria Adriana Cocozza, Jia Cheng Yuan, Caterina De Benedittis, Dajana Cuicchi, Alessandra Guido, Fabiola Lorena Rojas Llimpe, Antonietta D’Errico, Andrea Ardizzoni, Gilberto Poggioli, Lidia Strigari, Alessio Giuseppe Morganti, Franco Bazzoli, Luigi Ricciardiello, Rita Golfieri, Alessandro Bevilacqua

**Affiliations:** 1Department of Radiology, IRCCS Azienda Ospedaliero-Universitaria di Bologna, Via Albertoni 15, 40138 Bologna, Italy; francesca.coppola@aosp.bo.it (F.C.); arrigo.cattabriga@studio.unibo.it (A.C.); mariaadriana.cocozza@studio.unibo.it (M.A.C.); jiacheng.yuan@studio.unibo.it (J.C.Y.); caterina.debenedittis@aosp.bo.it (C.D.B.); rita.golfieri@unibo.it (R.G.); 2SIRM Foundation, Italian Society of Medical and Interventional Radiology, Via della Signora 2, 20122 Milan, Italy; 3Advanced Research Center on Electronic Systems (ARCES), University of Bologna, Via Toffano 2/2, 40125 Bologna, Italy; margherita.mottola@unibo.it (M.M.); alessandro.bevilacqua@unibo.it (A.B.); 4Medical and Surgical Department of Digestive, Hepatic and Endocrine-Metabolic Diseases, IRCCS Azienda Ospedaliero-Universitaria di Bologna, Via Massarenti 9, 40138 Bologna, Italy; dajana.cuicchi@aosp.bo.it (D.C.); gilberto.poggioli@unibo.it (G.P.); 5Department of Radiation Oncology, IRCCS Azienda Ospedaliero-Universitaria di Bologna, Via Massarenti 9, 40138 Bologna, Italy; alessandra.guido@aosp.bo.it (A.G.); alessio.morganti2@unibo.it (A.G.M.); 6Division of Medical Oncology, IRCCS Azienda Ospedaliero-Universitaria di Bologna, Via Albertoni 15, 40138 Bologna, Italy; fabiolalorena.rojas@aosp.bo.it (F.L.R.L.); andrea.ardizzoni2@unibo.it (A.A.); 7Pathology Unit, IRCCS Azienda Ospedaliero-Universitaria di Bologna, Via Massarenti 9, 40138 Bologna, Italy; antonietta.derrico@unibo.it; 8Department of Medical Physics, IRCCS Azienda Ospedaliero-Universitaria di Bologna, Via Massarenti 9, S. Orsola-Malpighi Hospital, 40138 Bologna, Italy; lidia.strigari@aosp.bo.it; 9Department of Medical and Surgical Sciences, IRCCS Azienda Ospedaliero-Universitaria di Bologna, Via Massarenti 9, 40138 Bologna, Italy; franco.bazzoli@unibo.it (F.B.); luigi.ricciardiello@unibo.it (L.R.); 10Department of Computer Science and Engineering, University of Bologna, Viale Risorgimento 2, 40136 Bologna, Italy

**Keywords:** rectal cancer, radiomics, MRI

## Abstract

Our study aimed to investigate whether radiomics on MRI sequences can differentiate responder (R) and non-responder (NR) patients based on the tumour regression grade (TRG) assigned after surgical resection in locally advanced rectal cancer (LARC) treated with neoadjuvant chemoradiotherapy (nCRT). Eighty-five patients undergoing primary staging with MRI were retrospectively evaluated, and 40 patients were finally selected. The ROIs were manually outlined in the tumour site on T2w sequences in the oblique-axial plane. Based on the TRG, patients were grouped as having either a complete or a partial response (TRG = (0,1), *n* = 15). NR patients had a minimal or poor nCRT response (TRG = (2,3), *n* = 25). Eighty-four local first-order radiomic features (RFs) were extracted from tumour ROIs. Only single RFs were investigated. Each feature was selected using univariate analysis guided by a one-tailed Wilcoxon rank-sum. ROC curve analysis was performed, using AUC computation and the Youden index (YI) for sensitivity and specificity. The RF measuring the heterogeneity of local skewness of T2w values from tumour ROIs differentiated Rs and NRs with a *p*-value ≈ 10^−5^; AUC = 0.90 (95%CI, 0.73–0.96); and YI = 0.68, corresponding to 80% sensitivity and 88% specificity. In conclusion, higher heterogeneity in skewness maps of the baseline tumour correlated with a greater benefit from nCRT.

## 1. Introduction

Colorectal cancer is the third most common cancer and the second leading cause of oncologic-related mortality in the world [1]. It is more common among men than women, although the incidence is rapidly increasing in the female population, and is 3–4 times more common in developed than in developing countries [2].

Locally advanced rectal cancer (LARC) is defined as a tumour penetrating through the entire bowel wall (stages 2, T3/T4N0) and/or having involvement of the regional lymph nodes (stage 3, any T N1/N2), and the standard of care treatment for patients with LARC currently involves neoadjuvant chemoradiotherapy (nCRT), followed by total mesorectal excision (TME) [3,4].

Together with endorectal ultrasound, magnetic resonance imaging plays an important role both in the primary staging and in the restaging of LARC after nCRT, according to the European Society for Medical Oncology (ESMO) guidelines [5,6]. Accurate imaging of the tumour and lymph nodes using high-quality MRI is essential in determining the local staging of rectal cancer, which is a critical marker when deciding whether to perform nCRT [6]. Recent evidence has suggested that 15–27% of patients will achieve a pathological complete response (pCR) to nCRT before surgery, suggesting that a “wait and watch” approach could be the best choice for these patients, thereby avoiding surgical complications. On the other hand, the percentage of patients who do not achieve tumour regression after nCRT, defined as non-responder (NR) patients, is reported to be between 7 and 30% [7,8].

In this scenario, early identification of NR and pCR patients before the beginning of nCRT could be of great value in developing a more tailored strategy of care, such as an intensified treatment regimen, as shown in PRODIGE23—a study that investigated the role of neoadjuvant mFOLFIRINOX before nCRT followed by TME surgery and adjuvant CT in resectable LARC [9].

Radiomics is the field of study exploiting machine learning on medical images, with the purpose of measuring visible properties or extracting more information than what the human eye can perceive [10,11]. It aims at extracting hidden imaging features from radiological images with the purpose of objectively and quantitatively describing tumour phenotypes and other pathological findings. Therefore, radiomic features (RFs) extracted from routinely acquired medical images might help with identifying predictive imaging biomarkers, thereby allowing the early detection of NR patients and patients who will obtain a pCR after nCRT [12,13,14,15,16].

The aim of this study was to assess the role of the RFs extracted from pre-therapy baseline T2-weighted (T2w) MR images in predicting the pathological responses of patients undergoing nCRT, thereby differentiating the group of patients with poor or minimal response from those obtaining a moderate response or a pCR, based on the tumour regression grade (TRG) assigned after surgical resection.

## 2. Materials and Methods

### 2.1. Study Population

A retrospective analysis of our electronic medical database for patients diagnosed with LARC who underwent MR examination for primary staging from January 2018 to December 2019 (total *n* = 85) was carried out.

The inclusion criteria were: (1) patients diagnosed with LARC in our institution undergoing (2) primary staging pretreatment MR, (3) treated with long-course neoadjuvant CRT followed by (4) TME. The exclusion criteria were: (1) patients who did not receive standard nCRT according to the directions of our multidisciplinary team, (2) patients who did not undergo surgical resection, (3) patients without available TRG information regarding the pathological report, (4) patients with mucinous adenocarcinoma hystotype and (5) patients who underwent MR examinations with incomplete staging MRI or imaging artefacts.

The Institutional Review Board approved the study and waived the requirement for written informed consent (number 842/2020/Oss/AOUBo).

### 2.2. Histopathological Reference Standard

The histopathological reports of the surgical resection samples at the time of TME were obtained for all the patients enrolled. Following the TRG staging system according to the American Joint Committee on Cancer (AJCC), the patients were initially divided into 4 groups [17]:-TRG0: no viable cancer cells (pathological complete response).-TRG1: single or small groups of tumour cells (moderate response).-TRG2: residual cancer outgrown by fibrosis (minimal response).-TRG3: minimal or no tumour cells killed (poor response).

For the purpose of the study, the patients were finally clustered into two groups, based on the TRG stage: (1) group one: patients with TRG = (0,1) (TRG0-1) who obtained a moderate response or a pCR; and (2) group two: patients with TRG = (2,3) (TRG2-3) who obtained a poor or minimal response.

### 2.3. Image Protocol Acquisition

All patients underwent pelvic MRI scans performed with 1.5 T MRI (GE Healthcare) according to the ESGAR guidelines for rectal cancer primary staging [18]. The protocol included three T2w fast spin echo (FSE) sequences: a sagittal high-resolution sequence, an oblique-axial high-resolution sequence orthogonal to the tumour axis and an oblique-coronal high-resolution sequence parallel to the tumour axis. Diffusion-weighted images (DWIs) were obtained in axial planes using echo-planar imaging (EPI) sequences at three *b*-values (b0, b600 and b1000 s/mm^2^), and restriction of diffusion was quantified by the apparent diffusion coefficient (ADC) value. In addition, a T1 axial sequence of the entire pelvis was acquired.

Bowel cleansing was performed with two days of a low-fibre diet and the oral administration of Macrogol-Na-K 14 gr the day before the study.

Technical data are reported in Table 1.

### 2.4. Tumour Segmentation

All oblique-axial high-resolution T2w FSE and DWI images were retrieved from the Picture Archiving and Communication System (PACS) (Carestream, Concord, ON, Canada) for image segmentation. A radiologist with more than 15 years of experience (FC) performed all the segmentations by manually outlining the lesion on each consecutive slide—excluding the intestinal lumen—using a designated, free, open source software package for visualisation and medical image processing (ImageJ, version 1.52a, available at https://imagej.nih.gov/ij, accessed on 9 November 2020) [19]. This was performed in two stages: (1) the entire tumour region was highlighted on 3 orthogonal planes of the space and (2) manual segmentation was performed on the 3 mm-thickness axial plane in all slices of the tumour site, by outlining a region of interest (ROI) for each of them. An example is shown in Figure 1. The radiologist was blinded to the histopathological results.

### 2.5. Radiomic Feature Generation

Eighty-four local first-order RFs were generated from T2w ROIs. In particular, seven local parametric maps were achieved for entropy, mean, median, skewness, kurtosis, interquartile range and coefficient of variation by investigating a square tissue unit of side length equal to 0.5 cm [20]. Twelve global histogram features were then computed on each of these seven maps, that is, maximum value, standard deviation, median absolute deviation, mean and median of the last decile in addition to the same seven descriptors introduced above for the local maps. The RFs were designed in order to measure the local variability of the MRI values in a simple and robust manner, with the aim of catching the local tissue heterogeneity, both within the tumour site, and between the tumour and its surrounding habitat. Figure 2 shows an example of the methodological workflow adopted, including image acquisition, LARC segmentation and RF extraction.

### 2.6. Statistical Analysis and Feature Selection

A univariate analysis was carried out on the original 84 RFs to select the one with the highest discriminative capability for the two groups TRG0-1 (true positives, TPs) and TRG2-3 (true negatives, TNs). It is well known that, to have the highest generalisation capability, the simplest discrimination model has to be adopted. In addition, the ratio r = N/l, with N being the number of samples of the smallest class and l the number of features finally selected, must be kept as high as possible. To that end, to prevent overfitting, only one RF was considered for discrimination [21]. A one-tailed Wilcoxon rank-sum test at α = 0.05 significance level was used to assess the statistical significance of the separation between the two groups. The overall selection procedure was carried out in three steps. First, RFs unable to statistically differentiate TRG0-1 and TRG2-3 were discarded. Second, the RFs surviving the first step were ranked according to their *p*-values after Bonferroni correction. Third, the feature showing the lowest *p*-value was selected and its discriminative capability was assessed using a ROC curve, by computing its area under the curve (AUC). To determine the best cut-off for the selected RF, the Youden index (YI) was computed and the values for specificity (SP) and sensitivity (SE) were computed accordingly. False positives (FPs), false negatives (FNs), positive predictive value (PPV) and negative predictive value (NPV) were also used to discuss performance. Separation between TRG0-1 and TRG2-3 was also assessed by computing median values and interquartile range (IQR) for each group.

## 3. Results

### 3.1. Clinical Characteristics

From the initial cohort of 85 patients, 11 patients who did not complete the standard nCRT, 24 patients who did not undergo surgical resection in our institution, 7 patients with no available TRG and 3 patients with incomplete MRI staging or MR images corrupted by artefacts were excluded (Figure 3).

Forty patients, 14 (35%) women and 26 (65%) men, having a median age of 65 years (interquartile range, IQR = 14.5 years) were finally included in the analysis (Table 2). According to the histopathological report, the 40 patients were split into two groups: 15 patients belonging to TRG0-1 and 25 patients belonging to TRG2-3. Due to the small number of patients enrolled, only a training dataset with the aim of a discrimination study was considered.

### 3.2. Discrimination Results

The TRG0-1 group of 15 patients, the smallest class and one feature finally having been selected, had a ratio of r = 15.

The RFs extracted from pre-therapy baseline MR images were analysed in order to differentiate TRG0-1 from TRG2-3 according to the AJCC classification, and we were able to discriminate different responses to subsequent nCRT.

Eighteen RFs came through the first feature selection step (*p*-value < 0.05); and s_e_, a local measurement of tumour heterogeneity, was the most discriminative RF to separate TRG0-1 from TRG2-3, with *p* ≈ 10^−5^—much lower than the significance threshold of α = 3∙10^−3^, considering the Bonferroni correction of α = 0.05. In particular, the RF selected was based on the parametric maps of local skewness, which were spread over a range of 2.82–5.26 for TRG0-1 and 2.88–8.74 for TRG2-3. To clarify, Figure 4a–d reports the skewness maps of four patients, having TRGs from 0 to 3, highlighting the evolution of local skewness variability.

In fact, while the asymmetries are localised in the maps of TRG0 and TRG1, as can be seen by the wide red and blue regions, these tend to shade in the skewness map of TRG2 and even more in TRG3, showing quite uniform speckles. Accordingly, the entropy computed on the skewness maps, s_e_, demonstrates the higher heterogeneity of TRG0 and TRG1 in Figure 4a,b, with respect to that of TRG2 and TRG3 in Figure 4c,d.

Figure 5 reports the ROC curve of s_e_, yielding an AUC = 0.90 (95% CI, 0.73–0.96), with a YI = 0.68 (cut-off s_e_ = 3.93), SE = PPV = 80% and SP = NPV = 88%.

Figure 6 shows the waterfall plot of s_e_ for the two groups, TRG0-1 (red bars) and TRG2-3 (blue bars), where the cut-off value was subtracted for visualisation purposes. Hence, the separation achieved at s_e_ = 0 highlighted three false positives and three false negatives.

The distance between the two groups can be better appreciated in the boxplots observed in Figure 7. Median values were s_e_-M = 4.08 (TRG0-1) and s_e_-M = 3.68 (TRG2-3) with IQR = 0.15[4.00, 4.15] and IQR = 0.46[3.40, 3.86] for TRG0-1 and TRG2-3, respectively.

## 4. Discussion

At present, there is growing interest by the medical community in radiomics applied to the study of colorectal cancer. For instance, using radiomic models, Liu et al. correlated the characteristics extracted from ADC maps with colorectal cancer stage and were able to highlight some RFs capable of differentiating tumour stages, thereby enabling them to predict the depth of local invasion (stage pT1-2 versus pT3-4) and possible lymph node involvement (pN0 versus pN1-2) [22]. Furthermore, Lu et al. used a radiomic approach to predict the histopathological prognostic factors of colorectal cancer, by differentiating stage pT1–pT2 tumours from stage pT3–pT4. Additionally, Huang et al. developed and validated a radiomic nomogram for the preoperative prediction of lymph node metastasis in colorectal cancer [23,24]. Nevertheless, there are a number of studies whose specific aim was the possibility of predicting the response to neoadjuvant therapy by means of radiomic analysis, which could prospectively tailor medical care based on tumour profiling [25].

To achieve this, T2-weighted images are considered the best choice in the evaluation and staging of LARC, as they offer valuable diagnostic performance [4]. However, some authors also included functional sequences, e.g., DW images. Liu et al. elaborated a radiomics model based on T2-weighted and DW images considering a cohort of 222 patients with LARC before and after nCRT, aiming to predict pathological complete response (AUC as high as 0.9756 in the validation cohort) [22,26]. Furthermore, through the segmentation of T2W and DW MR images, Bulens et al. designed models based on LASSO regression analysis, allowing a noninvasive prediction of response to nCRT in a cohort of patients with similar characteristics, thereby achieving a higher AUC (between 0.83 and 0.86) for predicting complete or near-complete pathological responses (ypT0-1N0), respectively [27].

It follows that the use of radiomics in the evaluation of LARC, as described above, might also allow identifying the responsive patients and providing them with targeted therapies while differentiating the non-responsive patients who could beneficiate of intensified neoadjuvant treatment regimen as neoadjuvant mFOLFIRINOX before nCRT [9]. More specifically, various authors have recently addressed the prediction of nCRT responses based on clinical assessments or different TRG staging systems [7,28,29,30,31,32,33,34]. However, the first relevant distinction of the present study is that in all those studies, the ratio r was far smaller than that in the present study. The highest ratio (r = 9.4) can be found in the study of Cusumano et al., which, however, achieved a worse separation (AUC = 0.77), and the remaining studies had ratios ranging from r = 5.5 to r = 0.87—namely, more features than patients [7,28]. Though their results were sometimes a bit better than ours (AUC = 0.92 [30] and AUC = 0.94 [28,31,33]), some others were much worse (AUC = 0.72 [7], AUC = 0.75 [34] and AUC = 0.82 [32]). Although the majority of these studies carried out an external validation of the predictive model on a holdout test-set, their very low r values made the generalisability of their models questionable.

In this study, a notable result was obtained through the combination of precision in the segmentation of the radiologists and the strict and accurate application of the principles of quantitative imaging by the engineers. The statistical analysis in this study yielded an excellent and significant (*p*~10^-5^) differentiation of patients based on their responses to the nCRT therapy (AUC = 0.90), starting from the s_e_ feature. The SE, SP, PPV and NPV represented a very good trade-off between the need for containing the risk of overtreatment and the detection of the responder patients. It goes without saying that the cut-off of s_e_ currently set at s_e_ =3.93 could be adjusted to ensure that no patient who needed nCRT was excluded from therapy, thereby ensuring a number of false negatives equal to 0 (i. e. 100% sensitivity). Consequently, this would lead to decreasing the specificity of the calculation and to increasing the number of false positives, with a consequent increase in nCRT-induced overtreatment. The two populations of TRG0-1 and TRG2-3 had very different characteristics which the radiomic analysis of the pre-therapy MR images allowed highlighting. In particular, the diversity between patients with different grades of nCRT response relied on tissue heterogeneity at baseline. In particular, the first group (TRG0-1) which identified patients who would respond completely to the therapy (TRG0) and those having single cells or small groups of tumour cells (TRG1), was characterised by a greater heterogeneity of skewness maps at baseline (s_e_-M = 4.08) with a very low variance. On the other hand, the second group (TRG2-3) of patients who would either respond poorly (TRG2) or show minimal response (TRG3) had a low heterogeneity of skewness maps at baseline (s_e_-M = 3.68); these values were also spread over a wider range. It is worth noting that this feature was not the heterogeneity of tumour tissue, but the heterogeneity of the skewness, computed on the local tissue regions. The skewness represents the asymmetry of local T2w values—namely, a departure from normality; and in case of an early alteration stage, it begins in the subregions of the ROI, each with a different evolving status (i.e., local skewness value), thereby yielding a heterogeneous picture of the ROI in TRG0-1. Instead, in a later evolution stage, the extension of the skewed regions dramatically shrinks, hinting at a greater similitude between the voxels of the ROIs, represented by a much less heterogeneous TRG2-3. Therefore, regarding the colormaps of TRG0-1, with clear red and blue regions highlighting the highest positive or negative skewness, it was not surprising that TRG0-1 lesions had the highest heterogeneity, and for the same reason, that the TRG2-3 lesions with their smoothed regions had the lowest.

The higher heterogeneity (contrast) in skewness maps in the baseline tumour has been correlated with a greater benefit from nCRT. Accordingly, the virtual curve resulting from the median values of s_e_ is a steep descending line and exactly indicates the course of the disease as expected. The present findings pointed out that radiomics could hopefully assist medical doctors in choosing the best therapy based on quantitative imaging data which, to date, have remained unnoted, and have never been considered in the choice of the optimal therapeutic process for the patient.

The pertinent literature illustrates the presence of many biomarkers that might correlate with the LARC response to nCRT (e.g., miRNAs). This could make it so that, in future works, the radiomics feature identified in this study (s_e_) can be correlated with these biomarkers to evaluate a possible implementation in clinical practice, allowing better adherence to patient tailored therapy [35,36,37].

A limitation of our study was the small number of patients enrolled, which allowed carrying out only a discriminatory study. Another was the retrospectiveness of the study. Validation with a prospective multicentre study is needed.

In conclusion, the results in this study highlighted a key clinical application of radiomics with the possibility of predicting nCRT response. The preliminary findings in this study which showed such a marked differentiation between TRG0-1 and TRG2-3 indicated that, in the future, patients could be stratified according to the four TRG stages. These data need to be confirmed in a large cohort of patients.

## 5. Conclusions

Patients with lower tumour heterogeneity of local skewness evaluated on MRI T2W-images at pre-therapy were those who had a better response to nCRT. The tumour heterogeneity of local skewness of pre-therapy patients showed a very promising role in predicting response to nCRT.

## Figures and Tables

**Figure 1 diagnostics-11-00795-f001:**
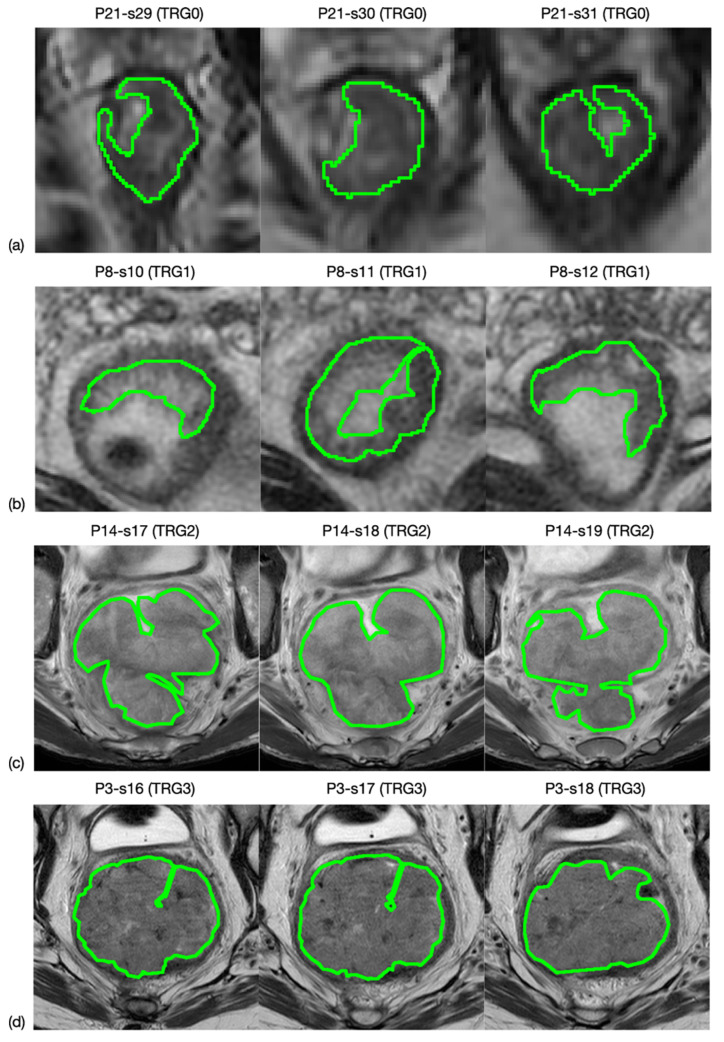
MRI image segmentation. Here is reported an example of how the segmentation process was carried out. In particular, line (**a**) shows three slices selected from an MRI study of a patient whose rectal cancer presented a TRG0 in the pathological study, while lines (**b**–**d**) correspond to TRG1, TRG2 and TRG3.

**Figure 2 diagnostics-11-00795-f002:**
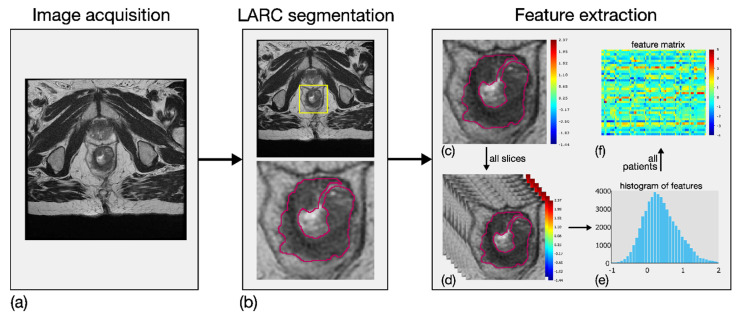
Methodological workflow adopted for the radiomic analysis, involving three main steps. First, image acquisition of MRI sequences (**a**). Second, outlining of LARC ROIs on all slices of the tumour site (**b**). Third, extraction of eighty-four RFs based on local first-order statistical descriptors. In particular, first-order features were computed locally in LARC ROI (**c**), for all the segmented slices (**d**), thereby achieving as many parametric maps. Twelve global descriptors were computed on the global histogram referring to each patient’s slices (**e**), and a feature matrix was achieved from all patients (**f**).

**Figure 3 diagnostics-11-00795-f003:**
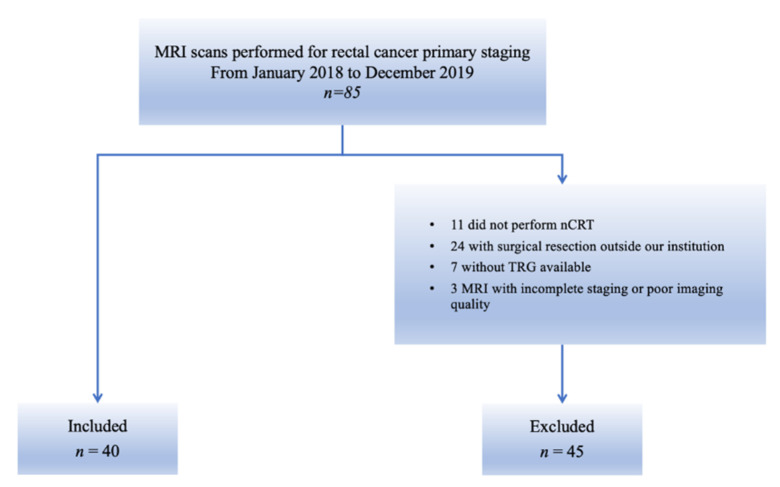
Flowchart of patient enrolment. The figure shows patients who were included in this study from the initial cohort and those who were excluded, according to the exclusion criteria.

**Figure 4 diagnostics-11-00795-f004:**
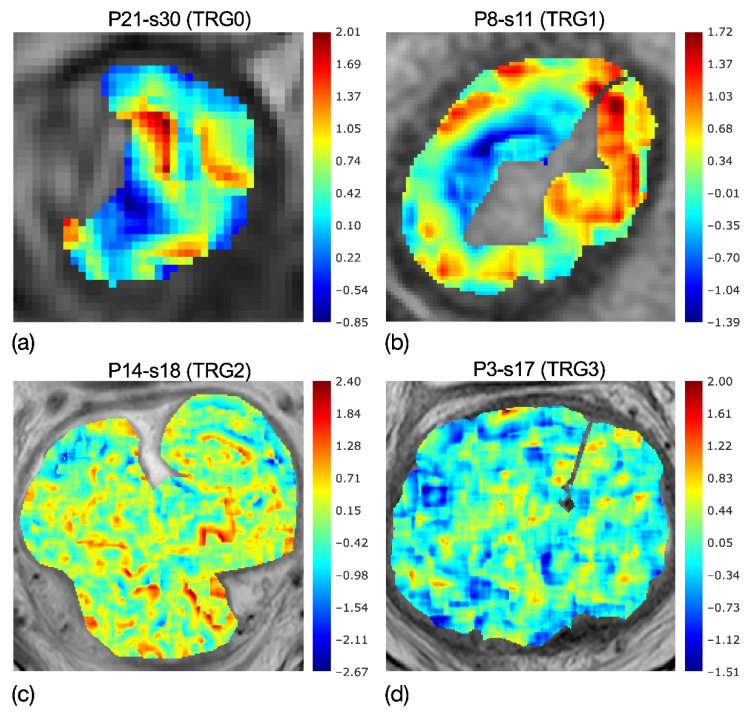
Colorimetric maps of local skewness of four patients with TRG0 (**a**), TRG1 (**b**), TRG2 (**c**) and TRG3 (**d**), highlighting the evolution of local skewness variability. While localised asymmetries are present in the maps of TRG0-1, quite uniform speckles are shown in TRG2-3. Accordingly, the entropy of the skewness catches the higher heterogeneity of (**a**,**b**) maps and the lower heterogeneity in (**c**,**d**) maps.

**Figure 5 diagnostics-11-00795-f005:**
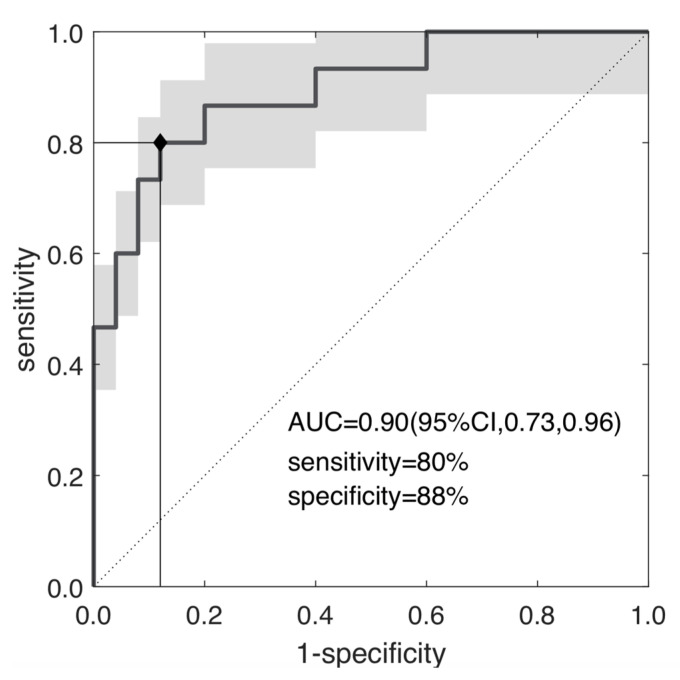
ROC curve of the selected radiomic feature; the skewness of the entropy; separating TRG0-1 and TRG2-3 with AUC = 0.90, sensibility = 80% and specificity = 88% at YI = 0.68 (*p*~10^−5^).

**Figure 6 diagnostics-11-00795-f006:**
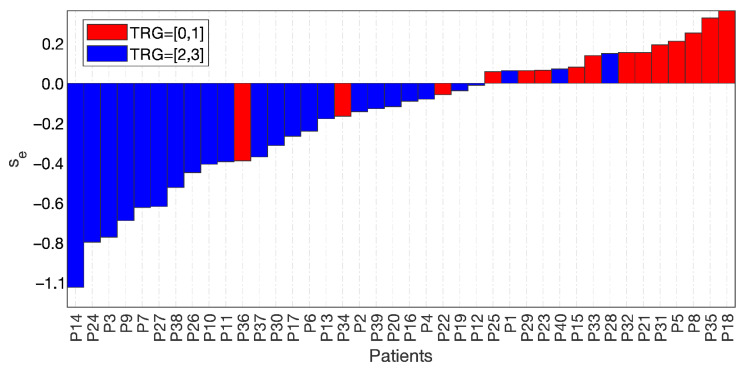
Waterfall plot of the two groups, TRG0-1 with red bars and TRG2-3 with blue bars, separated at a value of the skewness of the entropy = 0, with 3 false positives and 3 false negatives.

**Figure 7 diagnostics-11-00795-f007:**
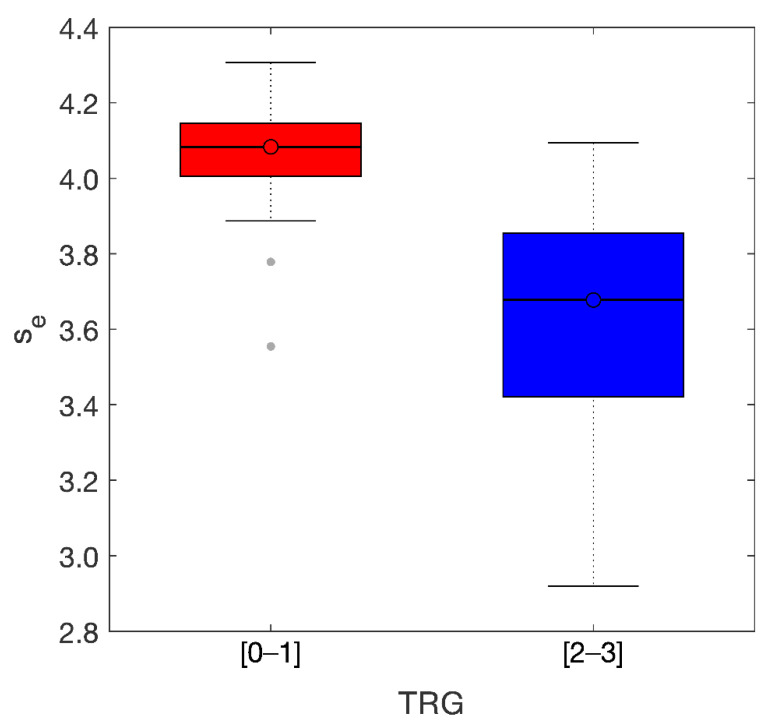
Boxplots of the two groups. Median values of the entropy of the skewness were 4.08 (TRG 01) and 3.68 (TRG 23) with IQR = 0.15 and IQR = 0.46, respectively.

**Table 1 diagnostics-11-00795-t001:** Technical data of the sequences acquired.

Sequence	TE (ms)	TR (ms)	Slice Thickness (mm)	Slice Spacing (mm)	FOV (cm^2^)
Sag T2 3 mm	110	2220	3,0	0	22
Ax T2 3 mm	110	2140	3,0	0	22
Cor T2 3 mm	110	2220	3,0	0	22
Ax e DWI b1000	79	5000	4,0	0	40
Ax T1 5 mm	9.6	580	5,0	0	32

**Table 2 diagnostics-11-00795-t002:** Baseline characteristics of the patients included in the analysis. ECOG-PS stands for “Eastern Cooperative Oncology Group performance status”; cT and cN stand for “clinical tumour stage” and “clinical node stage” according to TNM staging of rectal cancer.

Characteristics	Responder	Non-Responder	**Total (*n* = 40)**
	TRG 0-1 (*n* = 15)	TRG 2-3 (*n* = 25)	
Sex, males/females, *n*%	11/4	16/9	25/15
Median age (range),	73.3/26.7	64.0/36.0	62.5/37.5
Years	66 (50–85)	60 (46–81)	61 (46–85)
ECOG-PS 0	13 (86.7%)	21 (84.0%)	34 (85.0%)
ECOG-PS 1	1 (6.7%)	2 (8.0%)	3 (7.5%)
ECOG-PS 2	1 (6.7%)	2 (8.0%)	3 (7.5%)
cT	T3 15 (100%)	T3 17 (68.0%)	T3 32 (80.0%)
	T4 0	T4 8 (32.0%)	T4 8 (20.0%)
cN	N − 5 (33.3%)	N − 5 (20.0%)	T3 10 (25.0%)
	N + 10 (66.7%)	N + 20 (80:0%)	T4 30 (75.0%)

TRG—tumour regression grade; ECOG-PS Eastern Cooperative Oncology Group performance status; cT clinical tumour stage and cN clinical node stage in according to TNM staging of rectal cancer.

## Data Availability

The data are not available because of patients’ privacy.

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
