# Peer review of "The Heterogeneity of Skewness in T2W-Based Radiomics Predicts the Response to Neoadjuvant Chemoradiotherapy in Locally Advanced Rectal Cancer"

_diagnostics, 2021, doi:10.3390/diagnostics11050795_

Round 1
Reviewer 1 Report
In this manuscript, the authors analyse a relevant question of if pre-therapy heterogeneity of local skewness in radiomics predicts the response to neoadjuvant chemoradiotherapy in locally advanced rectal cancer.
Comment 1: Figure 1 is insufficiently labelled and insufficiently described in the figure legend. Figure 1 should be a supplementary figure. Figure 2 has a title but no legend. Figure 3, 4 and 5 legends contain unnecessary acronyms – ‘se’ and ‘FN/FP’ – while it is clear, these should be spelt out. All figures legends should be able to be read independent of the results text.
Comment 2: This study is interesting and provides novel data. Patient cohorts are well-described, the sample size is small, though further explanation should be provided on samples, i.e., a more expansive table of baseline characteristics of the two cohorts should be included.
Comment 3: For Figure 3, the more pixelated appearance of maps a and b (TRG 0 and 1) may indicate imaging of a smaller region, has smaller tumour length also been considered as a factor? Please include more images of pre-therapy baseline MR images so that three can be viewed per TRG.
Comment 4: Expand the discussion - the authors should suggest possible reasons why there is skewness heterogeneity as it appears that they only speculate that it’s not ‘the heterogeneity of tumour tissue,’. Include more rationale from literature on why tumours have different levels of radiosensitivity and that a number of cellular processes and immune mechanisms have been implicated in radioresponse phenotypes
Minor points / formatting
Editing needs to be done in terms of incorrect use of commas instead of full stops, e.g., line 59, 279 and throughout.
Paragraphs are short – combine sentences into a paragraph and a bit more background in introduction and discussion as relevant.
Swapping between TRG X with a space and TRGX without a space – there should be a space, and other inconsistencies.
Unnecessary use of acronyms makes the text unclear, e.g., FNs 266 and FPs 268.
Referencing is sometimes in an incorrect positioning – place citation number at the end of the sentence.
258, reword to remove 'thanks to'
Author Response
Dear Revewer,
Thank you for giving us the chance to improve on our work titled: “The heterogeneity of skewness in T2W-based radiomics predicts the response to neoadjuvant chemoradiotherapy in locally advanced rectal cancer”. We especially appreciate your valuable comments, as they gave us the opportunity to better clarify some aspects of our paper. Below you can find each of your questions, followed by our response, and the relative section of the paper.
Comment 1: Figure 1 is insufficiently labelled and insufficiently described in the figure legend. Figure 1 should be a supplementary figure. Figure 2 has a title but no legend. Figure 3, 4 and 5 legends contain unnecessary acronyms – ‘se’ and ‘FN/FP’ – while it is clear, these should be spelt out. All figures legends should be able to be read independent of the results text.
We modified figure 1, labelling it and enriching its legend. We added a legend to figure 2 too. Lastly, we removed the unnecessary acronyms, as you suggested.
Comment 2: This study is interesting and provides novel data. Patient cohorts are well-described, the sample size is small, though further explanation should be provided on samples, i.e., a more expansive table of baseline characteristics of the two cohorts should be included.
Thank you for stressing this important point. As you suggested we added a new table (Table 2) which summarizes the baseline characteristics of the two cohorts.
Comment 3: For Figure 3, the more pixelated appearance of maps a and b (TRG 0 and 1) may indicate imaging of a smaller region, has smaller tumour length also been considered as a factor? Please include more images of pre-therapy baseline MR images so that three can be viewed per TRG.
Thank you for pointing this out. As you recommended, we added more images of pre-therapy baseline MRI, for each TRG (now Figure 1).
Comment 4: Expand the discussion - the authors should suggest possible reasons why there is skewness heterogeneity as it appears that they only speculate that it’s not ‘the heterogeneity of tumour tissue,’. Include more rationale from literature on why tumours have different levels of radiosensitivity and that a number of cellular processes and immune mechanisms have been implicated in radioresponse phenotypes
We revised the text accordingly. We added a paragraph in the last part of the discussion in order to better explain the role of this radiomics feature and its possible correlation with other biomarkers in the evaluation of the response to nCRT (line 328).
Minor points / formatting
Editing needs to be done in terms of incorrect use of commas instead of full stops, e.g., line 59, 279 and throughout.
Paragraphs are short – combine sentences into a paragraph and a bit more background in introduction and discussion as relevant.
Swapping between TRG X with a space and TRGX without a space – there should be a space, and other inconsistencies.
Unnecessary use of acronyms makes the text unclear, e.g., FNs 266 and FPs 268.
Referencing is sometimes in an incorrect positioning – place citation number at the end of the sentence.
258, reword to remove 'thanks to'
We changed the text according to your comments.
Reviewer 2 Report
Thank you for the opportunity to review this interesting paper. The authors have performed a modern T2-texture analysis to predict treatment Response of rectal cancer in a neoadjuvant setting.
The paper is well written and the topic is timely.
The patient sample is rather large for a single center study.
You could expand the discussion on studies, which performed a direct correlation between T2-histogram analysis and histopathology in rectal cancer as well as add a part regarding DWI.
Author Response
Dear Revewer,
Thank you for giving us the chance to improve on our work titled: “The heterogeneity of skewness in T2W-based radiomics predicts the response to neoadjuvant chemoradiotherapy in locally advanced rectal cancer”. We especially appreciate your valuable comment, as it gives us the opportunity to better clarify some aspects of our paper. Below you can find your comment, followed by our response, and the relative section of the paper.
Comment: You could expand the discussion on studies, which performed a direct correlation between T2-histogram analysis and histopathology in rectal cancer as well as add a part regarding DWI.
We sincerely appreciated your comment and we modified our manuscript accordingly, adding a section in the “discussion” paragraph (line 264).